# The Manic Idea Creator? A Review and Meta-Analysis of the Relationship between Bipolar Disorder and Creative Cognitive Potential

**DOI:** 10.3390/ijerph20136264

**Published:** 2023-06-30

**Authors:** Boris Forthmann, Karin Kaczykowski, Mathias Benedek, Heinz Holling

**Affiliations:** 1Institute of Psychology, University of Münster, 48149 Münster, Germany; 2Institute of Psychology, University of Graz, 8010 Graz, Austria

**Keywords:** bipolar disorder, creativity, divergent thinking, creative cognitive potential, meta-analysis

## Abstract

Even though a relationship between psychopathology and creativity has been postulated since the time of ancient Greece, systematic meta-analyses on this topic are still scarce. Thus, the meta-analysis described here can be considered the first to date that specifically focuses on the relationship between creative potential, as measured by divergent thinking, and bipolar disorder, as opposed to psychopathology in general. An extensive literature search of 4670 screened hits identified 13 suitable studies, including a total of 42 effect sizes and 1857 participants. The random-effects model showed an overall significant, positive, yet diminutively small effect (*d* = 0.11, 95% CI: [0.002, 0.209], *p* = 0.045) between divergent thinking and bipolar disorder. A handful of moderators were examined, which revealed a significant moderating effect for bipolar status, as either euthymic (*d* = 0.14, *p* = 0.043), subclinical (*d* = 0.17, *p* = 0.001), manic (*d* = 0.25, *p* = 0.097), or depressed (*d* = −0.51, *p* < 0.001). However, moderator analyses should be treated with caution because of the observed confounding of moderators. Finally, none of the employed methods for publication-bias detection revealed any evidence for publication bias. We discuss further results, especially regarding the differences between subclinical and clinical samples.

## 1. Introduction

Since the time of the ancient Greece philosophers, the idea of the “mad genius” has insinuated a connection between psychopathology and creativity [1]; for most laypeople, this image still exists today [2]. This putative association between psychopathology and creativity has also been the subject of empirical psychological research. However, most studies on this topic have examined the link between creativity and psychopathology in general rather than focusing on the link between creativity and a specific disorder [3]. Furthermore, studies have not conceptualized creativity in a homogeneous way, as most have focused on creative achievement, but others on creative professions, or creative personality, and only a few measured creativity with established assessments of creative cognitive potential [4].

In addition, studies that have focused on the specific link between bipolar disorder and creative potential, so far, have revealed inconsistent findings: while some studies found a positive relation between bipolar disorder and creative potential [5,6], some found no relation between the two constructs [7,8], and other studies even indicated a negative relationship between bipolar disorder and creative potential [9]. To clarify the relationship between bipolar disorder and divergent thinking as the most widely used indicator of creative potential, this meta-analysis considered all relevant data and took relevant moderators into account that might explain these contradicting results.

### 1.1. Creativity, Creative Potential, and Divergent Thinking

Since creativity is multi-layered and a complex concept, definitions of *creativity* usually focus on specific aspects of the construct. In the standard definition of creativity, Runco and Jaeger [10] defined creativity based on two criteria: originality and effectiveness. Runco [11] has further differentiated between creative performance and creative potential. Creative performance is what laypeople generally understand as creativity: a creative profession, creative achievement, even reaching the level of a creative genius, as exemplified by famous creative individuals (e.g., Vincent van Gogh or Virginia Woolf). Additionally, creative performance can further be separated into different areas of performance, such as artistic creativity in the fields of art, music, and literature, or scientific creativity in scientific fields [12]. However, because not all people have (yet) reached the point of actually behaving or performing creatively, one must also take into account their potential for creative ideas. Creative (cognitive) potential refers to a person’s potential to produce original and effective ideas that may lead to creative performance [11]. Creative potential is also conceptually related to everyday creativity (i.e., little-c creativity) [13], which is one of the most important domains for investigating creativity in the general population. Hence, in this meta-analysis we aimed at studying divergent thinking tasks as the most prominent assessments of creative potential [14] and everyday creativity [13].

Divergent thinking (DT; [14]) tasks are open-ended tasks that require participants to generate ideas. DT performance can be scored for fluency, flexibility, originality, and elaboration [15,16,17]. Fluency refers to the number of different ideas or solutions generated to address a specific problem; flexibility typically refers to the number of different conceptual categories from which ideas were generated; originality can be defined as remoteness, novelty, and the cleverness of ideas [18]; and elaboration describes the detailedness of produced ideas [15,16,17]. These facets of DT are not independent but are intertwined. For example, generating a greater number of ideas (i.e., greater fluency) enhances the likelihood of generating more original ideas [19]. Moreover, the relationship between facets depends on how scores are aggregated across responses: Summative scorings, which add scores across all responses, result in large correlations between fluency and other facets (i.e., fluency confound [20,21,22]), whereas discriminant validity of facets can be maintained when scorings are independent of fluency. This can be achieved by dividing the number of original ideas by the total number of ideas (i.e., average scoring [23,24]), or by scoring subsets of ideas rather than scoring every idea individually as in the snapshot scoring [23,25] or top scoring [26,27]. Common examples of DT tasks include the Alternate Uses Test (AUT; [28]) and the Torrance Test of Creative Thinking (TTCT; [29,30]). These tests further intend to measure DT on various modalities such as the verbal, figural, or numeric content modality.

In addition, time-on-task has been shown to influence performance in DT tasks [31]. More lenient time limits imply more responses as well as more original responses. This can be explained by the serial order effect which refers to the observation that quality of responses is a function of time during testing [32,33]. Typically, high-quality responses are generated at later stages of the idea generation process. In addition, rather strict time limits may introduce task speededness which potentially affects the measured construct [34,35]. Another relevant factor for assessing creativity relates to the type of instruction participants are given to perform the DT task, since participants can be instructed to generate as many ideas as possible, only original ideas, or only appropriate ideas [36,37,38]. Different instructions, therefore, emphasise different DT facets, either stressing the *quantity* or *quality* of ideas, which influences DT performance [39].

As demonstrated above, assessing divergent thinking is complex and involves a variety of specifications. For that reason, we will include moderator variables around the assessment of DT, such as the type of DT task (e.g., AUT, TTCT, or other.), DT facet (fluency, flexibility, originality, elaboration, or total), modality domain (figural, verbal, numeric, or mixed), time-on-task, method of aggregation (summed, averaged, or other), and instruction (be-fluent or be-creative). Notably, previous meta-analyses focusing on divergent thinking have emphasised that such assessment-related moderators are essential to understand heterogeneous effect sizes in the literature [40,41].

### 1.2. Bipolar Disorder

Bipolar Disorder (BD), also known as manic–depressive disorder, is a mood disorder involving extreme mood swings between depressive and elevated moods [42]. BD is more multifaceted than a simple mood disorder and is considered a “bridge between the two diagnostic classes [schizophrenia and depressive disorders] in terms of symptomatology, family history, and genetics” ([42], p. 123). Further, bipolar and related disorders are generally referred to in the plural rather than in the singular since their presentations are manifold. To be precise, the fifth edition of the Diagnostic and Statistical Manual of Mental Disorders (DSM-5) differentiates between bipolar I, bipolar II, and cyclothymic disorder, varying in different lengths and intensities of affective episodes [42]. Regarding affective episodes, one can distinguish between depressive, manic, and hypomanic episodes. Furthermore, individuals with bipolar I or II disorder experience phases without a (hypo)manic or depressive episode and, in between episodes, return to their euthymic baseline level of mood [42].

A depressive episode is characterized by a distinct period of at least two weeks with either depressed mood and/or loss of interest, pleasure, or both, along with accompanying symptoms [42]. Manic and hypomanic episodes can both be described as “a distinct period of abnormally and persistently elevated, expansive, or irritable mood” ([42], p. 124), in which at least three (or four, if the mood is not expansive but only irritable) of seven criteria must be met (for a full description of diagnostic criteria, see [42]). To name some examples, typical behaviors during a (hypo)manic episode are, for instance, seeking high-risk situations such as disproportional speeding in traffic, irresponsible sexual contacts, and unaccountable financial expenses [42,43]. A manic episode has to last at least one week, while a hypomanic episode lasts at least four days; and while a manic episode usually comes along with severe social or occupational impairment, such as hospitalization, harm to self or others, or psychotic symptoms, the functioning level in a hypomanic episode is not severe enough for hospitalization [42].

To fulfill the diagnostic criteria for bipolar I disorder, the criteria for a manic episode must be fulfilled at least once during a person’s lifetime (bipolar I, criterion A [42]), independent of the prior existence of a depressive episode. To fulfill the diagnostic criteria for bipolar II disorder, the criteria for at least one hypomanic episode as well as for at least one depressive episode must be met during a person’s lifetime (bipolar II, criterion A [42]), without ever meeting the criteria for a manic episode (bipolar II, criterion B [42]). For both, bipolar I and bipolar II, the (hypo)manic and/or depressive episode(s) cannot be better explained by other disorders, such as schizoaffective, schizophreniform, delusional, or other specified or unspecified disorders from the schizophrenia spectrum. While the manic episodes often cause the most significant impairment in individuals with bipolar I disorder, for individuals with bipolar II, the depressive episodes or the unexpected switches between episodes are usually stated as the most constricting, and they would rarely consider hypomanic symptoms as a reason for treatment [42].

Taking all various types of bipolar and related disorders together, the lifetime-prevalence can be estimated at around 3.9–4.4% [42,44]. As in most psychiatric diagnoses, risk factors are not univocal but originate from a combination of biological, social, and psychological factors [42,45]. However, family history of BD seems to be one of the greatest risk factors, resulting in a 10- to 15-times higher risk of falling ill with BD if relatives are known to suffer from it [42,46].

Interestingly, a paragraph about the relationship between BD and creativity can be found in the *DSM-5*:
“There may be heightened levels of creativity in some individuals with a bipolar disorder. However, that relationship may be nonlinear; that is, greater lifetime creative accomplishments have been associated with milder forms of bipolar disorder, and higher creativity has been found in unaffected family members. The individual’s attachment to heightened creativity during hypomanic episodes may contribute to ambivalence about seeking treatment or undermine adherence to treatment”.([42], p. 136)

Notably, this is the only paragraph in the whole *DSM-5* that deals with the topic of creativity, thus hinting at BD’s special significance in this regard. Additionally, as some bipolar patients might not undergo treatment due to the fear of, in a sense, losing their heightened creativity, the nature of this relationship must be clarified, which will help to reduce prejudice towards treatment and encourage everybody who needs treatment to seek it.

When taking a closer look at the specifications of BD, the complexity of the disorder as well as the influencing factors become evident. For that reason, the meta-analysis conducted here aimed to include all the variables intertwined with BD. To cope with the diversity of specifications around BD, the moderators examined in this meta-analysis include the specific type of BD (e.g., bipolar I, mixed sample of bipolar I and bipolar II, or subclinical BD), current episode (euthymic, depressed, hypomanic, or manic), diagnostic assessment of the disorder, comorbidities, duration, onset, hospitalization, number of episodes in the past, as well as medication.

### 1.3. Theoretical Framework for a Relationship between Creativity and Bipolar Disorder

#### 1.3.1. Mad Genius Theory

When examining the biographies of famous creative individuals from different creative fields such as the arts, music, or science, the mad genius link also seems to be ubiquitous. Examples such as Vincent van Gogh, who, presumably suffering from BD, cut off his ear and eventually committed suicide; Robert Schumann, who created 12.3 hymns on average during his manic episodes, while on average creating only 2.7 during his depressive episodes [47]; or John Forbes Nash, one of the most important mathematicians in the 20th century, who suffered from paranoid schizophrenia [48]. Self-evidently, many famous and genius artists, musicians, and scientists do not or did not suffer from mental illness—yet these examples do not seem to win over the media. Stories of the suffering artist or the crazy scientist are often portrayed in books, films, and TV series [49,50], popularizing the idea of the mad genius link in the general public.

Psychological research has contributed empirical evidence to inform the debate. One of the first studies to examine overall mental illness in a creative sample, also known as the landmark study, was conducted in 1987 by Andreasen [51], stating that in the study, reportedly 80% of creative writers (compared to 30% of nonwriters) suffered from a mood disorder. Even greater public attention was given to Jamison’s book Touched with Fire in 1993 [52], in which the author examined mental illness in a sample of people in creative fields such as poets, writers, and visual artists, stating that the probability of mental illness in creative individuals was 30 times higher than in non-creative people [52]. Following this path, in The Price of Greatness, Ludwig [53] reviewed 1004 biographies of individuals published in the New York Times between 1960 and 1990, most of which were in creative professions such as art, music, and poetry, concluded that great creativity comes at the price of madness.

All studies by these authors, namely Andreasen, Jamison, and Ludwig, are often cited as evidence for a mad genius link, not only in psychological research but also in popular media. However, these studies have been harshly criticized, especially by Schlesinger in 2009, pointing to poor methodological quality [54]. First, the participants in these studies were not diagnosed with mental disorders; instead, they simply self-reported their mental well-being. Furthermore, most of the studies did not have a control group, which is why no in-depth statistical analyses could be conducted and only percentages of mental illness are stated. On top of that, there is the long duration over which data was collected, the small number of participants, as well as a personal connection between the participants and the researchers.

Overall, the evidence from psychological research for a mad genius link is subject to debate and still based heavily on anecdotal rather than empirical grounds. Although the connection between psychopathology and creativity is not well supported, the idea that mental illness coincides with enhanced levels of creativity seems to be something considered as fact by the general population. Even though the stereotype of the mad genius may not be empirically true, they still ‘form a reality in people’s minds and shape how they perceive and behave towards gifted individuals’ ([55], p. 1).

#### 1.3.2. Shared Phenotype

Even though there is no clear support of the mad genius theory by scientific research, it makes sense to explore a possible connection between BD and creative ideation based upon shared composition. To begin with, typical symptoms of BD and the abilities needed in creative thinking may be related, especially to the following symptoms that appear during a (hypo)manic episode: racing thoughts, distractibility, and increased goal-directed activity (criterion B 4–6 [42]). Hence, Jamison [52] drew a connection between hypomanic episodes and creative potential, saying that the
“[t]wo aspects of thinking in particular are pronounced during creative and hypomanic thought—fluency, rapidity, and flexibility of thought, on the one hand, and the ability to combine ideas or categories of thought in order to form new and original connections, on the other. […] Speed per se, that is, the quantity of thoughts and associations produced in a given period of time, may be enhanced. The increased quantity and speed of thoughts may exert an effect on the qualitative aspects of thought as well; that is, the sheer volume of thought can produce unique ideas and associations.”([52], p. 105)

To extend this idea, a (hypo)manic episode might increase fluency, which increases the likelihood of generating original ideas. Indeed, this link between quantity (fluency) and quality (originality) in idea generation is implied by Simonton’s equal odds baseline [56,57,58] for the number of original ideas and also by the dual pathway model of creativity for originality on average [59].

Furthermore, Murray and Johnson [60] developed a schematic representation between the BD phenotype and creativity. They distinguish between generativity/novelty and consolidation/usefulness. Generativity relates to the production of various ideas (i.e., fluency) and consolidation relates to editing and polishing of ideas, mostly related to originality. The core idea in their theory is that elevated positive affect during mania might lead to greater DT, topped up by impulsivity, which might boost ideational processes without being inhibited by self-censorship [60]. In this scheme, a number of variables overlap either between BD and generativity, and/or BD and consolidation. The three-way overlap between BD, generativity, and consolidation includes the following variables: positive affectivity, openness, extraversion (performance), neuroticism (arts), neuroticism (science), high goal setting, romantic aesthetic, familial creativity, and evening preference (arts), which all have a positive direction. However, the direction of the association is more ambiguous for the two two-way relationships. The overlap between BD and generativity is assumed to go along with psychoticism, impulsivity, loose associations, and imagery, which are all associated positively, whereas the overlap between BD and consolidation has mixed associations. Impulsivity, intrinsic motivation, substance misuse, and anxiety disorder have a negative connotation, and only the drive to achieve has a positive association.

In conclusion, there is a variety of behavioral, cognitive, and motivational variables that BD and the two facets of creativity have in common. However, the scheme portrays a rather flat net of connectivity; for instance, neglecting dynamic relationships such as the influence of depressive episodes in BD.

#### 1.3.3. Shared Vulnerability

Carson [1] suggested a model of shared vulnerability, postulating that there are overlapping vulnerability factors between creativity and psychopathology that, in combination with either protective or risk factors, result in either enhanced creativity or psychopathology (psychopathology here refers to BD, schizophrenia, and substance abuse only). Derived from the fact that both creativity and BD are heritable [61,62,63] and to some extent polygenetic [64,65], the idea of a genetic linkage stands to reason. Indeed, polygenetic risk scores for BD were found to predict creative professions to a modest degree [66]. As such, Carson [1] generally assumes that creativity and psychopathology are similar in that they give individuals enhanced access to subconscious material. While the risk factors for psychopathology (low intelligence, working memory deficits, and perseveration) lead to being overpowered by this unconscious material, resulting in pathological symptoms, the protective factors (high intelligence, working memory skills, and cognitive flexibility) ensure the necessary meta-cognitive control to turn one’s bizarre and unusual thoughts into a creative advantage.

The three vulnerability factors identified by Carson [1] that may either result in enhanced creativity or psychopathology (depending on protective versus risk factors) are attenuated latent inhibition, preference for novelty, and hyperconnectivity. First, latent inhibition (LI) refers to how well sensory stimuli are selected and filtered [67,68]. Having a low LI implies that irrelevant information enters consciousness more easily [1,69]. A low LI has been found not only in individuals with psychosis (mainly schizophrenia but also BD; [70,71]) but also in individuals with high openness to experience, which, in turn, is the personality trait that most consistently predicts creative potential [72,73,74]. Furthermore, a low LI can be induced by psychoactive substances, which again have been linked to elevated levels of creativity [75] (but also see [76]).

Second, the factor preference for novelty or novelty seeking is an intrinsic motivational variable of heightened interest for exploring and approaching new stimuli [77]. Novelty seeking is related to both creativity and BD, since it was found that creative individuals prefer novel and complex stimuli over familiar and simple stimuli [78,79], and people with BD were found to show higher novelty seeking, especially during episodes of mania or hypomania [80].

Finally, the factor hyperconnectivity, described as abnormal neural linkage between brain areas that typically have no functional connection [1], has been detected both in people with mental disorders and in highly creative people performing a creative task [81,82,83]. Further, hyperconnectivity is a common phenomenon in people with synesthesia, namely cross-modal sensory associations such as connecting colors with musical tones, which is prevalent more often in creative individuals [84]. As speculated by Ramachandran and Hubbard [85], patterns of hyperconnectivity may even be associated with human metaphorical thinking, a thinking style often seen in creative people as well as people experiencing hypomania, psychotic episodes, or drug intoxication [1,85].

Taken together, these shared vulnerability factors offer theoretical grounds to assume a link between BD and creativity. Yet, the model does not imply that bipolar individuals are more or less creative; it only indicates that BD and creativity share certain dispositions, but it depends on yet other (protective versus risk) factors to eventually enhance either BD or creative thinking, but not necessarily both. As such, the meta-analysis at hand will try to incorporate variables that act as either protective factors or risk factors and, thus, seeks to include intelligence as a moderator in the analysis.

#### 1.3.4. Inverted U-Shaped Relationship

One of the most prevalent types of relationship is the idea of an inverted U-shaped relationship between BD and creativity [86,87,88]. Here, a curvilinear relationship between DT and BD is hypothesized, such that DT performance is expected to peak at a certain, rather subclinical level of BD, whereas individuals without any symptoms of BD as well as severely ill individuals should show lower performance [86,88]. Greenwood [89] also postulated an inverted U-shaped model, in which creative thinking and other desirable traits are expected to be enhanced by genetic loading, but at some point they cross over into undesirable regions as signs of psychopathology appear. This non-linear relationship is further foreshadowed by the above quote from the *DSM-5* on the relationship between DT and BD, saying that greater creative accomplishments are related to milder forms of BD and that creativity might be enhanced in healthy relatives of people suffering from BD.

Following the hypothesis of an inverted U-shaped relationship, it might also be interesting to study the relationship between DT and BD in the context of subclinical expressions of BD. Hence, studies using a subclinical sample will also be included in this meta-analysis and examined as moderator. One of the most commonly used diagnostic tools for assessing subclinical symptoms of BD is the Hypomanic Personality Scale (HPS; [90]), which was found to correlate highly with hypomanic symptoms [91]. Hence, to more accurately map the different levels of impairment due to BD, this meta-analysis will include severity as a moderator, thus allowing us to investigate a possible inverted U-shaped relationship.

### 1.4. Previous Findings on the Link between Bipolar Disorder and DT

As stated before, BD seems to hold a special position in the relationship between creativity and psychopathology, as seen in DSM-5’s paragraph specially dedicated to creativity as being related to BD [42]. Regardless of its presumable significance, once again empirical studies on the relationship between BD and creativity are rare, and have focused heavily on creative achievement, creative professions, or creative personalities rather than creative potential. To find out whether individuals with BD are creative, the most obvious approach would be to simply ask them about their self-observed creative actions during affective episodes. This is exactly what McCraw and colleagues [92] did in their study with a sample of 213 patients diagnosed with either bipolar I or II disorder. Results showed that, indeed, 82% of patients confirmed that they were creative during hypomanic and manic episodes. A subsequent qualitative analysis on a subsample of the patients showed that bipolar patients mostly engaged in writing, painting, work, or generating business ideas. However, the study did not include a control group, so the results only indicate that bipolar patients show creative actions during (hypo)mania; no proposition could be made as to whether bipolar patients show more creative behavior compared to healthy individuals. Other research has studied the relationship between BD and types of occupations [93], the relationship between BD and the Barron–Welsh Art Scale [6,94], the relationship between risk for BD and measures of creative achievements or convergent creative thinking [95], for example.

Focusing on studies that used DT as a measure of creativity to examine the relationship between creativity and BD in clinical and subclinical samples, the results appear to be just as diverse. For instance, Siegel and Bugg [7] found no significant relationship between the risk for hypomania as measured by the HPS and DT as measured by the ATTA. In fact, none of the four subscales (fluency, flexibility, originality, and elaboration) were associated with measures of hypomania risk [7]. On the other hand, other studies such as the study by Alici et al. [9] showed a negative relationship between BD and DT. In their study, remitted patients with BD were compared to healthy controls, whereby BD patients showed significantly worse performance in fluency and originality in AUT. In contrast, other studies have shown increased DT performance in bipolar patients: in a study by Srivastava and colleagues [96], for example, bipolar patients scored better in verbal and figural DT tasks than healthy controls and depressed patients. In line with this, Claridge and Blakey [97] found that hyperthymic and cyclothymic temperament could predict heightened performance in DT tasks. In summary, the body of empirical studies on the relationship between BD and DT is limited, and existing studies show diverging results, calling for a systematic meta-analytic examination of the average effect and potential moderators.

### 1.5. Previous Meta-Analytic Findings

A recent meta-analysis by Paek and colleagues [98], which focused on an overall mad genius connection by investigating the relationship between three common psychopathologies—ADHD, anxiety, and depression—and little-c creativity, did not find a substantial connection between both (overall correlation: *r* = 0.06). However, when taking a closer look, differences in the strength of the relationship were found between the three psychopathologies, and ADHD and creativity even had a negative association (*r* = −0.17). This highlights the relevance of examining different disorders separately. So far, only two meta-analyses have explicitly included BD: first, the systematic review and meta-analysis by Taylor [99], which examined the connection between creativity and mood disorders, and, second, the meta-analysis by Baas and colleagues [100], which contrasted creativity in approach-based vs. avoidance-based psychopathologies, including the *risk* for BD in the cluster of approach-based psychopathologies.

The systematic review and meta-analysis by Taylor [99] examined the connection between creativity and mood disorders, including unipolar depression, dysthymia, and BD. Three separate meta-analyses were conducted to understand the direction of the relationship, investigating (1) mood disorders in creative individuals, (2) creativity in individuals with mood disorders, and (3) a covariance of creativity and mood disorders. The first group of effect sizes, focusing on mood disorders in creative individuals, included 1805 participants from 10 studies and 24 effect sizes, and found a moderate to large effect (*g* = 0.64). The second analysis, focusing on creativity in people with mood disorders, included 8,316,598 participants from 13 studies and 76 effect sizes and yielded a small and not statistically significant connection (*g* = 0.08). The third analysis on the covariance of both included 3965 participants from 15 studies and 59 effect sizes and suggested a small yet significant correlation of *r* = 0.09. Taken together, the results by Taylor [99] point towards the idea that creative people in fact show a greater prevalence for having a mood disorder than non-creative people, while people with a mood disorder do not seem to be more creative in general.

Even though this study gives insightful results—not only regarding a connection between mood disorders and creativity but also regarding the direction of the effect—there are still some important limitations. First and foremost, the effect sizes were not calculated for specific disorders but as an overall effect for mood disorders. When taking a closer look at the results from the moderation analysis, especially from the first meta-analysis, it is noticeable that the level of creativity differs between the different types of disorders: while dysthymia is not linked to creativity at all across all analyses, BD seems to have the strongest connection, which might have biased the strength of the overall effect, once again making it clear that disorders should be analysed individually. Moreover, Taylor [99] used several indicators of creativity, ranging from everyday creativity over creative performance to creative achievement, such that DT was only one of many variables used to operationalize creativity. For instance, DT and creative accomplishments only have a small (*r* = 0.22) correlation [101,102], which is not sufficient to merge these variables into an overall creativity score. Therefore, the current meta-analysis solely focuses on DT, because it is the most commonly used indicator of creative potential [14].

Furthermore, calculating three separate analyses to investigate the direction of the effect is an interesting and insightful approach, but it results in a smaller number of studies per analysis. Rather than conducting three separate analyses with smaller sample sizes over various disorders, the current meta-analysis rather conducts one meta-analysis that only focuses on BD and incorporates the assessment method as a moderator together with other relevant factors.

The meta-analysis by Baas et al. [100] also included BD when investigating creativity but took a slightly different approach. Instead of focusing on a clinical sample, Baas and colleagues investigated the vulnerability or risk for psychopathologies in non-clinical samples. Additionally, the relationship between creativity and psychopathology was not investigated by specific disorders but was separated into two clusters based on motivational approach and avoidance systems: approach-based psychopathologies included positive schizotypy and risk for BD, and avoidance-based psychopathologies included anxiety, negative schizotypy, and depressive mood. Results showed that approach-based psychopathologies were indeed related to higher creativity (positive small effect), while avoidance-based psychopathologies were related to lower creativity. Taking a closer look, results also showed that risk for BD was associated with higher creativity (*k* = 28; *r* = 0.22), indicating that people with a greater risk of suffering from BD are also more creative. Yet again, these results highlight the need for a disorder-specific approach regarding the connection between creativity and psychopathology, but they still must be taken with caution. First and foremost, Baas et al. explicitly examined a non-clinical sample, whereas the connection between creativity and BD in the whole spectrum of clinical and subclinical samples is relevant. Therefore, rather than looking at bipolar risk via family history or related diagnostic assessment methods such as temperament, as in Baas et al. [100], the current meta-analysis focuses on clinically diagnosed BD, including presentations of this diagnosis as euthymic, depressed, or manic, as well as subclinical BD. Furthermore, just as in the meta-analysis by Taylor [99], Baas et al. [100] used many variables to operationalize creativity, roughly separating them into clusters of creative performance (DT is included here) and self-rating of creativity (in terms of personality and creative behavior). Apparently, DT was defined rather broadly, since the Remote Associates Test or tasks of verbal fluency were also included in the category of DT; however, the Remote Associates Test corresponds to convergent thinking, and verbal fluency cannot be used synonymously with fluency in view of Guilford’s classification [15,16], which means these measures are not equivalent to DT. To avoid ambiguity in the conceptualization of creativity, the current meta-analysis solely focuses on creative potential measured by DT tasks and excludes studies that made exclusive use of such other measures.

### 1.6. Aims of the Present Study

Considering all the different presentations of BD, BD is not only a common mental illness [42,44], but, due to its associated high suicide rates, it is also one of the deadliest [42,103]. Therefore, more in-depth research is needed on the protective factors and risk factors of BD in order to build a better understanding of the disorder and continuously work on better treatment methods. Since it is still debated whether creativity is unrelated, positively related, or negatively related to BD, this meta-analysis aims to offer systematic research on variables associated with BD. Further, this meta-analysis aims to challenge myth-based conceptions about the relationship between psychopathology and creativity that appear to be driven by anecdotal evidence. The idea of a mad genius often romanticizes people struggling with mental illness, making it sound desirable to have a mental disorder in order to benefit from the creative genius that comes along with it. On the other hand, mental disorders are still stigmatized in today’s society [104], such that considerable negative consequences come along with these diagnoses. To neither romanticize a mental illness nor to deny possible positive implications, the present meta-analysis aims to assemble empirical facts to enable a better understanding of the connection between BD and creative thinking.

To do so, this meta-analysis makes use of distinct definitions and assessments of both BD and creativity. As opposed to existing meta-analyses that have commonly focused on psychopathology as a whole, the current meta-analysis solely focuses on BD, which arguably shows the strongest theoretical and empirical relationship to creativity [1,60]. Specifically, in comparison to the meta-analysis by Taylor [99], the current study only examines BD rather than a whole cluster of mood disorders and, different from the meta-analysis by Baas et al. [100], the current meta-analysis focuses on clinical and subclinical BD rather than just the risk for BD.

On top of that, the present meta-analysis focuses on creative potential as assessed with DT tasks, since we were interested in everyday, little-c creativity rather than Pro-C or Big-C creativity, and aimed to avoid broad, ambiguous conceptualizations of creativity (e.g., creative achievement, professions, personality, or convergent thinking).

To identify factors that affect the relationship between BD and creative potential, the current meta-analysis also considers relevant moderators. They include general moderators on study characteristics such as study age (i.e., 2019 minus year of publication), study country, participants’ mean age, use of a matched control group, as well as study design (correlational vs. mean differences). Due to the diverse assessment of creative potential, it also investigates moderators related to the kind of DT assessment including type of DT test, scoring of DT test, domain of DT test, instruction, and calculation method. As stated above, BD is a complex psychopathology, which is why the current study also analyses moderators on specific presentation of BD, status of BD, BD assessment tool, duration, onset, severity, medication, and comorbidities.

Finally, we evaluated moderators for their potential confounding which can undermine straightforward interpretations of moderator findings [105], assessed if any single studies or effect sizes had a particularly strong influence on the findings, and examined publication bias.

## 2. Method

### 2.1. Study Collection

#### 2.1.1. Inclusion and Exclusion Criteria

Studies were included in this meta-analysis if the following criteria were met: (a) The study assessed creative potential in terms of little-c creativity by using measurements of DT (e.g., AUT, TTCT, or ATTA). Studies measuring creativity on the Pro-C or Big-C level, such as focusing on creative achievement, or creative profession were excluded. Similarly, studies using other measures of creative potential than DT, such as verbal fluency or convergent creative thinking, were also excluded. (b) The study used established measurements to assess clinical or subclinical BD (e.g., assessment by a clinician, diagnostic interviews, or questionnaires especially designed to measure symptoms of BD). Studies only using self-reported psychopathology without a prior diagnosis were excluded. (c) The study had to include quantifiable measures of creativity and psychopathology. Therefore, theoretical reviews, case reports, and similar works were excluded. (d) Studies comparing mean differences had to have a healthy control group (CG). Studies only comparing two pathological groups or only using a preselected group as CG (e.g., creative controls) were excluded. For correlational studies, no CG was needed. (e) The study either had to report an effect size or had to report sufficient statistical information to calculate an effect size. In studies comparing mean differences, at least the means and standard deviations had to be reported. Correlational studies had to report the *r*-value for zero-order bivariate relationships. The reported effect size had to be convertible to the effect size used in the study, which is Cohen’s *d*; (f) the study had to focus primarily on BD. Because the majority of patients suffering from BD also have at least one other comorbid diagnosis [42,106], studies also incorporating patients with comorbidities were not automatically excluded. However, BD had to be the main diagnosis; studies with a sample suffering primarily from another psychological or physiological condition were therefore excluded. (g) The study had to be conducted on a human sample; animal studies were excluded. (h) In the case that the same sample was used in several studies, only the published version or the more substantial version was used. (i) The study had to be published in a language known to the authors (German, English, Polish, or Norwegian).

#### 2.1.2. Literature Search

A literature search for potentially relevant studies was conducted using the electronic databases PsychINFO, PSYNDEX, and Medline. The following search term was used for PsychINFO and Medline: (bipolar* OR mood disorder OR manic* OR mania OR cyclothym* OR hyperthym*) AND ((divergent think* OR creative think* OR creativ* OR convergent think* OR little-c OR fluency OR flexibility OR originality) OR (alternate use* OR masselon* OR insight* OR remote associat* OR torrance test)) and was adjusted for German language within PSYNDEX. Notably, the search term was more general, as implied by the relationship between BD and DT (i.e., it includes convergent think*, insight*, or remote associate*). This choice is based on the fact that sometimes studies with a focus on a different measure of creative thinking (e.g., the Remote Associates Test) may also assess a measure of DT as a complementary variable. Such studies can still be eligible for a meta-analysis and, hence, the search term was in accordance with a comprehensive search strategy. There were no restrictions regarding date or publication location.

Ultimately, 5989 articles were found via database searching (as of 10 July 2019). Further, other resources, such as forward- and backward-searches on the basis of the two meta-analyses by Baas et al. [100] and Taylor [99], as well as a keyword search in Google Scholar, yielded another *m* = 59 studies. After removal of duplicates, a total of *m* = 4670 studies were screened for eligibility. On the basis of the title and abstract, *m* = 4541 studies were excluded for not fitting the subject matter. Hereupon, *m* = 129 full-text articles were further screened for eligibility, applying the inclusion and exclusion criteria mentioned above. As a result, *m* = 13 studies, with 13 samples and 42 effect sizes were included in this meta-analysis. The references for all included studies can be found in the OSF repository.

Figure 1 shows a flowchart with more detailed information on the inclusion process. For simplicity, an excluded study was only listed in one of the exclusion categories, even though multiple reasons for exclusion may have applied. Therefore, the number of studies in each exclusion category is only estimated. Some studies fit inclusion criteria and collected data on DT and BD but unfortunately did not report an effect size or necessary data to calculate an effect size; the authors of these studies were contacted via e-mail and asked whether it was possible to obtain the effect size or data needed to calculate an effect size. If the authors did not answer within the course of one week, a reminder e-mail was sent, including a time limit of two weeks to deliver effect sizes in order to be included in this meta-analysis. Of the authors contacted, three were able to obtain the missing effect size(s), so their studies were included in this meta-analysis. Two authors reported that the information of interest was no longer accessible, and one author did not reply. The literature search is fully documented in a file that is openly available in the Open Science Repository (https://osf.io/tnksa/; accessed on 12 June 2023).

### 2.2. Coding Procedure

The coding procedure was developed and defined by communication between the first, second, and fourth author of this work. The first author coded all the variables alone but discussed coding decisions with other members of the department in case of uncertainty or inconsistent information. Detailed information on the coding scheme can be found in Table A1 in Appendix A. A detailed description of the coding process can be found in the OSF repository (https://osf.io/tnksa/; accessed on 12 June 2023).

### 2.3. Extraction of Effect Sizes and Model Fitting

The current meta-analysis was conducted using the statistics program *R* (Version 3.1.3; [108]) using the extension package *metafor* (Version 2.1; [109]). The analysis script and the data are openly available in the project’s OSF repository (https://osf.io/tnksa/; accessed on 12 June 2023). First, we provide descriptive statistics of the coded effect sizes. Second, we report the quantified overall effect size. Third, we report moderator analyses in the following order for different sets of moderators: (a) study and sample characteristics, (b) DT-related moderators, and (c) BD-related moderators. Finally, we assessed bivariate associations between moderators to reveal any potential confounding effects among moderators, examined if any effect sizes represented influential cases, and assessed publication bias.

Even though most studies were based on correlational data (correlational: *m* = 9, mean differences: *m* = 5), more effect sizes were based on mean differences (correlational: *k* = 19; mean differences: *k* = 23), which is why the effect size used for this meta-analysis was Cohen’s *d*. In studies calculating mean differences, Cohen’s *d* was calculated based on the means and standard deviations reported in the studies. In correlational studies, Pearson’s *r* was converted into Cohen’s *d* by using the formula $d=\frac{2r}{\sqrt{1-r^{2}}}$, as introduced by Borenstein and colleagues [110], with the variances being calculated by $V_{d}=\frac{4V_{r}}{{\left(1-r^{2}\right)}^{3}}$, with $V_{r}=\frac{{\left(1-r^{2}\right)}^{2}}{n-1}$ [110].

Due to the nature of the studies being conducted in different locations and by different researchers, the idea of a random-effects model as opposed to a fixed-effects model seemed plausible [110]. To see whether using one source of variance was already sufficient compared to a multilevel model [111] that used both sources of variance, such as between-study and within-study variances, likelihood ratio tests were conducted. Since the between-study variance was estimated to be zero, the full model using both sources of variance did not uncover any additional information on variance compared to the model using only the within-study variance ($\chi^{2}$ = 0.00, *p* = 1.00). Thus, a model accounting for the between-study variance was not needed but the within-study variance was necessary ($\chi^{2}$ = 17.16, *p* < 0.001). Therefore, all further analyses used the simple random-effects model, not the multilevel model using both sources of variance.

Importantly, some studies reported an additional total creativity score besides the individual scorings for fluency, flexibility, originality, and elaboration. Since the total score might be confounded with the individual scorings [22], it might be inappropriate to include the total scores in the analysis if the total score was not the only scoring used. To control for the possible confounding, analyses were also calculated with a subset of effect sizes (*m* = 13, *k* = 37) excluding total scores whenever other scores were also reported (*k* = 5 excluded). Results only differed slightly (confidence intervals were strongly overlapping) between the entire group of effect sizes (*d* = 0.11, 95% CI: [0.00, 0.21], *p* = 0.045) and the subset of effect sizes without the additional total scores (*d* = 0.09, 95% CI: [−0.02, 0.20], *p* = 0.124). Thus, for clarity and simplicity, we used the entire number of extracted effect sizes in all the following analyses.

## 3. Results

In total, this meta-analysis included 1857 participants from 13 studies with 42 effect sizes. The mean age of participants was 28.01 years (*SD* = 7.54, range = 16.06–38.06) with an average of 60.11% female participants (*SD* = 7.01, range = 47.66–72.22). On average, there were 142.85 participants per study (*SD* = 98.06, range = 58–437) across all included studies. Taken together, 9 studies (69.23% of studies) with 19 effect sizes (45.24% of effect sizes) used a correlational design, while 5 studies (38.46% of studies) with 23 effect sizes (54.76%) used mean differences. Here, the total percentage of studies equals more than 100%, since one study reported both correlational effect sizes and effect sizes for mean differences on the basis of different assessment tools. Of the 13 studies, only 1 study (7.69% of studies) with two effect sizes (4.76% of effect sizes) was unpublished; all other studies were published.

Regarding the assessment tools for creativity, 5 studies (38.46% of studies) with 12 effect sizes (28.57% of effect sizes) used tools based on Torrance’s TTCT, 7 studies (53.85% of studies) with 14 effect sizes (33.33% of effect sizes) used tools based on Guilford’s AUT, and 1 study (7.96% of studies) with 16 effect sizes (38.10% of effect sizes) used other tools. Namely, the study used the inventiveness scale from the Berlin Intelligence Structure Test [112], which, next to an AUT task, also includes several other tasks. The study also included a task, where participants had to give explanations for a specific scenario, and a *masselon task*, where participants had to form sentences from three given words; this is why the study could not be coded as only AUT. Regarding the assessment of BD, 5 studies (38.46% of studies) with 23 effect sizes (54.76% of effect sizes) used tools based on the *DSM*, 6 studies (46.15% of studies) with 12 effect sizes (28.57% of effect sizes) used the HPS, and 3 studies (23.08% of studies) with 7 effect sizes (16.66% of effect sizes) used other measures. More detailed information on study, sample, and effect sizes, including moderators, and *Q*-statistics, can be found in Table A4 in Appendix B.

### 3.1. Overall Effect Size on the Relationship between BD and DT

In total, 42 effect sizes were included in this meta-analysis. Of these, 30 (71.43%) were positive and 12 (28.57%) were negative. A histogram for the distribution of effect sizes is shown in Figure 2. The overall effect across all studies in the random-effects model was significantly positive yet diminutively small (*d* = 0.11, 95% CI: [0.00, 0.21], *p* = 0.045). A forest plot for the overall effect can be found in Figure 3.

Since the main fundamental difference between the included studies lies in the two different study types, we conducted further subset meta-analyses for study type (mean differences vs. correlational). In all subclinical samples correlational data was used, while clinical samples reported data based on mean differences; only one clinical study reported both mean differences and correlational data. The random-effect meta-analysis for the subset of studies using mean differences was not significant (*d* = 0.05, *k* = 23, 95% CI: [−0.13, 0.22], *p* = 0.602), but the subset meta-analysis for studies using correlations showed a significant positive result (*d* = 0.18, *k* = 19, 95% CI: [0.08, 0.27], *p* < 0.001). Forest plots for the correlational and mean differences subsets can be found in the Appendix (see Figure A1 in Appendix B for correlational and Figure A2 in Appendix B for mean difference forest plots).

The conducted overall random-effects model indicated considerable heterogeneity (*Q* = 116.57, *df* = 41, *p* < 0.001, $I^{2}$ = 67.13%), which means that random variance does significantly differ from zero. To account for the found heterogeneity, moderator analyses were conducted on the level of study and sample characteristics as well as coded variables regarding DT and BD.

### 3.2. Analysis of the Moderation Effects by Study and Sample Characteristics

#### 3.2.1. Study Age

All studies were from 2006 (13 years old) or newer, except two studies published in 1990 (29 years old) and 1997 (22 years old). Study age did not significantly account for any amount of the overall effect size (*Q* = 1.02, *df* = 1, *p* = 0.313, *τ* = 0.28).

#### 3.2.2. Country

There was a moderation effect of the country the study was conducted in that was significant by trend (*Q* = 7.75, *df* = 3, *p* = 0.051, *τ* = 0.26), such that one study from Turkey ended up having a large influence (*d* = −0.97; 95% CI: [−1.75, −0.19]; *p* = 0.015). Contrast analyses were only conducted between country presentations with at least four effect sizes, namely between the USA and Poland. The contrast between studies from those countries was not significant.

#### 3.2.3. Study Type

The type of study did not account for significant variance in the overall effect size (*Q* = 1.28, *df* = 1, *p* = 0.259, *τ* = 0.27); yet, there was a significant, positive effect for correlational studies (*d* = 0.17, 95% CI: [0.02, 0.32]; *p* = 0.027), but no significant effect for difference studies (*d* = 0.05, 95% CI: [−0.09, 0.19]; *p* = 0.503).

#### 3.2.4. Mean Age

The mean age of participants did not explain significant variance of the overall effect size (*Q* = 2.88, *df* = 1, *p* = 0.090, *τ* = 0.31). One study did not report the mean age of participants.

#### 3.2.5. Age Group

The age group of participants had no significant effect on the overall variance in effects (*Q* = 1.64, *df* = 3, *p* = 0.651, *τ* = 0.28). Yet, it is important to note that most participants were classified as adults (*m* = 5, *k* = 25) or young adults (*m* = 6, *k* = 15), while only one study with one effect size was classified as being mixed and including school children.

#### 3.2.6. Gender

There was no significant effect of gender ratio on the overall variance of effect sizes (*Q* = 0.78, *df* = 1, *p* = 0.377, *τ* = 0.29). One study with two effect sizes did not report gender information.

#### 3.2.7. Matched CG

Whether or not the CG was matched did not explain significant variance of the overall effect size (*Q* = 0.01, *df* = 1, *p* = 0.910, *τ* = 0.38). Notably, of the studies having a CG, only two used a matched CG; three studies used no matched CG.

### 3.3. Analysis of the Moderation Effects Regarding DT

#### 3.3.1. DT Test

Categorisation of DT tests as either AUT, TTCT, or other did not have a significant moderating effect on the overall variance of the effect size (*Q* = 0.27, *df* = 2, *p* = 0.872, *τ* = 0.28). When examining the DT measures, namely either AUT (*m* = 5, *k* = 8), UUT (*m* = 2, *k* = 6), ATTA (*m* = 3, *k* = 8), TTCT-F (*m* = 1, *k* = 2), TTCT-V (*m* = 1, *k* = 2), or BIS (*m* = 1, *k* = 16), no significant effect on the overall variance of effect sizes could be found either (*Q* = 4.78, *df* = 5, *p* = 0.443, *τ* = 0.28). Interestingly, all effects were positive except for the effect of TTCT-F (*d* = −0.37, *p* = 0.117).

#### 3.3.2. DT Scoring

The scoring of DT tests used, namely either fluency, flexibility, originality, elaboration, or total, did not explain any variance in effect sizes (*Q* = 1.07, *df* = 4, *p* = 0.898, *τ* = 0.30). The direction of the effect for flexibility (*d* = −0.05, *p* = 0.889) was negative, while it was positive for fluency (*d* = 0.09, *p* = 0.302), originality (*d* = 0.05, *p* = 0.628), elaboration (*d* = 0.03, *p* = 0.933), and total (*d* = 0.18, *p* = 0.068). Once again, it is important to mention that flexibility and elaboration each only comprised one effect size from one study.

#### 3.3.3. DT Domain

Whether the domain of the DT test was verbal, figural, mixed, or numeric had no significant influence on the variance of the effect size (*Q* = 5.12, *df* = 3, *p* = 0.162, *τ* = 0.26). Interestingly, the direction of the effect differed between domains: while the figural (*d* = −0.12, *p* = 0.390) and numeric domains (*d* = −0.07, *p* = 0.705) had a negative effect size estimate, the verbal (*d* = 0.14, *p* = 0.056) and mixed domains (*d* = 0.22, *p* = 0.026) had a positive effect size estimate (verbal by trend). Notably though, the two domains with a negative connection contained a small number of studies and effect sizes (figural: *k* = 3, *m* = 6; numeric: *k* = 1, *m* = 4). Contrast analyses showed a significant contrast between the figural and mixed domain (*d* = −0.33, *p* = 0.047), but this no longer remained significant after Holm-adjusting the *p*-value for multiple testing (*p* = 0.283).

#### 3.3.4. DT Time

The time to perform the DT test had no significant moderating effect (*Q* = 0.10, *df* = 1, *p* = 0.748, *τ* = 0.28).

#### 3.3.5. DT Instruction

Instruction classified as either quantity or quality of ideas did not significantly affect the overall variance of effects (*Q* = 0.07, *df* = 1, *p* = 0.789, *τ* = 0.28).

#### 3.3.6. DT Calculation

The calculation of the DT test as either summed, averaged, or other did not significantly explain any variance in the overall effect size (*Q* = 1.48, *df* = 2, *p* = 0.477, *τ* = 0.28). However, the majority of studies and effect sizes reported summed scores (*k* = 8, *m* = 33), while only a handful of studies reported either averaged (*k* = 4, *m* = 7) or other (*k* = 2, *m* = 2) scorings. One study with one effect size did not report information on how DT scores were calculated, and it was not obtainable through other context information regarding the specific DT test.

### 3.4. Analysis of the Moderation Effects Regarding BD

#### 3.4.1. BD Presentation

Presentation of BD as bipolar I, bipolar mixed, or subclinical did not show a significant moderating effect (*Q* = 1.10, *df* = 2, *p* = 0.576, *τ* = 0.28). The effect for the subclinical presentation was significant (*d* = 0.17, *p* = 0.039), but not for the clinical presentations.

#### 3.4.2. BD Status

The BD status showed a significant moderating effect (*Q* = 18.86, *df* = 3, *p* < 0.001, *τ* = 0.20). While a depressed status had a significant, negative effect (*d* = −0.51, *p* < 0.001), a status of euthymic (*d* = 0.14, *p* = 0.043) or subclinical (*d* = 0.17, *p* = 0.001) had a significantly positive effect. A manic status had a positive yet not significant effect (*d* = 0.25, *p* = 0.097). Significant contrasts were found between depressed and all other status presentations (depressed and euthymic: *d* = −0.65, *p* < 0.001; depressed and subclinical: *d* = −0.68, *p* < 0.001; depressed and manic: *d* = −0.76, *p* < 0.001) that remained significant after Holm-adjusting for the *p*-value (*p* < 0.001; *p* < 0.001, *p* = 0.001). Importantly, all of the four effect sizes for depressed and for manic statuses stemmed from the same study.

#### 3.4.3. BD Assessment

The tool of assessment for BD showed no significant effect on the variance of the overall effect size (*Q* = 1.77, *df* = 2, *p* = 0.413, *τ* = 0.28). Interestingly, assessments using the HPS yielded a significant positive effect (*d* = 0.21, *p* = 0.029).

#### 3.4.4. Clinical Diagnosis

BD being diagnosed by a professional clinician (either fully, partly, or not at all) did not have a moderating effect on the overall variance of the effect size (*Q* = 1.86, *df* = 2, *p* = 0.394, *τ* = 0.28). However, only no clinical diagnosis was significant (*d* = 0.17, *p* = 0.036).

#### 3.4.5. Comorbidity

The existence of comorbidities had no moderating effect on the overall effect (*Q* = 0.10, *df* = 1, *p* = 0.755, *τ* = 0.35). However, in total, many studies (*k* = 8, *m* = 17) did not report whether participants had comorbidities; of these, two were clinical studies with four effect sizes, and the rest were subclinical studies.

#### 3.4.6. Medication

Whether all or some participants in the sample took medication for BD had no significant moderating effect (*Q* = 0.84, *df* = 1, *p* = 0.360, *τ* = 0.37). Importantly, the group of all participants taking medication only consisted of 1 study with 16 effect sizes, while the mixed group was made up of 4 studies with 9 effect sizes. Overall, 6 studies (13 effect sizes) did not report any information on the use of medication for BD.

### 3.5. Associations between the Moderators

For an overview about frequencies of all characteristics of moderators, see Table A5 for categorial moderators and Table A6 for continuous moderators in Appendix B. To examine possible confounding between moderators, their associations were also computed. As indicated by Lipsey [114], confounding between moderators in meta-analyses is rather common, since moderator variables of interest are frequently related. Therefore, it is important to also take confounding of moderators into account to make more precise and non-misleading interpretations. In the current meta-analysis, moderators of different levels of measurements were used, which is why interrelationships were calculated using Pearson’s correlations, point biserial correlations, Eta (η), and Cramer’s *V*. Pearson’s correlations were used for interrelationships between continuous moderator variables; point biserial correlations were used for interrelationships between continuous and categorial (dichotomous) variables; η was used for continuous and categorial (polytomous) variables; and Cramer’s *V* was used for interrelationships between categorial variables.

As anticipated, associations were found between most of the moderators. See Table A7 in Appendix B for an overview of all interrelationships between moderators. However, because of missing data and uneven distributions, not all associations between moderators were computed. For these reasons, interrelationships also have to be interpreted with care. Regarding the bivariate relationships, most of the results in the above-mentioned moderator analyses should be handled with care, especially those showing numerous highly significant intercorrelations, such as country, study type, mean age, age group, DT test, DT time, BD presentation, BD assessment, and clinical diagnosis.

### 3.6. Influencing Studies

To see whether any studies or effect sizes had particularly high influence, a Baujat plot [115] for the random-effects meta-analysis was conducted; it can be found in Figure 4. In the Baujat plot, three effect sizes from two studies seem to be influential. These three effect sizes were negative and relatively high regarding absolute values (*d* = −0.97; *d* = −0.77; *d* = −0.64). However, an outlier analysis did not show a significant influence of these three or any other effect sizes. Nonetheless, to control for the possible influence of these three effect sizes apparent in the Baujat plot, an overall effect size was computed once again without the potentially influential effect sizes. When excluding the potentially influential effect sizes from the overall analysis, only small changes in the overall effect size could be observed (*d* = 0.16, 95% CI: [0.08, 0.15], *p* < 0.001, Δ*d* with-without influential effect sizes = 0.05), self-evidently resulting in a larger effect size. Nevertheless, the non-significant outlier analysis as well as the results from the adapted random-effects meta-analysis indicate that the influential studies can be included in further analyses, since the difference in effect size due their exclusion was negligible.

### 3.7. Publication Bias

To test the occurrence of a publication bias, we conducted a funnel plot, trim-and-fill-test, Egger’s test, and rank correlation test. Visual analysis reveals symmetry of the funnel plot, as can be seen in Figure 5. The trim-and-fill test also indicated no missing studies on the left side. Further, the moderator analysis of publication status as either published or unpublished yielded no significant difference in publication status (*Q* = 4.25, *df* = 2, *p* = 0.119, *τ* = 0.28). However, only one unpublished study was included, and most unpublished studies were not obtainable within the time this meta-analysis was conducted. In line with this, the Egger’s test (*z* = −1.51, *p* = 0.132) as well as the rank correlation test (*τ* = −0.09, *p* = 0.458) also indicated no publication bias.

In addition, Rosenthal’s Fail-safe *N* was calculated to see how many additional studies would be needed to reduce the observed significant effect (*p* < 0.001) to a target significant effect (*p* = 0.05), indicating that 167 studies would be needed. As introduced by Mullen and colleagues [116], the *Fail-safe ratio* calculated by $N/\left(5\times k+10\right)$, in which *k* refers to the number of experiments or studies, is a useful post hoc rule of thumb to interpret the Fail-safe *N*.

If the ratio score exceeds 1, it can be assumed that the weight of the included studies is tolerant for future results [116]. In this case $167/\left(5\times 13+10\right)=2.23$ is larger than 1, indicating that this meta-analysis is robust against a publication bias.

When examining the appearance of the funnel plot for the whole sample of effect sizes, it seems evident that most of the standard errors are placed on few vertical lines within the plot. This might be a result of the use of different scorings for DT tasks; as such, we conducted more in-depth analyses for the individual subsets of scoring methods for a possible publication bias. Regarding the scoring, a publication bias for fluency (*z* = −0.67, *p* = 0.504; *τ* = −0.11, *p* = 0.591) seems unlikely. However, for originality (*z* = −2.67, *p* = 0.008; *τ* = −0.55, *p* = 0.026), the Egger’s test and the rank test indicate that a publication bias cannot be ruled out, though the trim-and-fill test shows that no study is missing on the left side of the funnel. Funnel plots for fluency and originality can be found in Figure A3 and Figure A4 in Appendix B. Estimates of publication biases for the scorings of flexibility and elaboration could not be computed due to the small number of effect sizes.

Estimates of publication biases were also conducted for the two subsets of the moderator study design. Publication biases seem unlikely for the two subsets including the mean difference subset (*z* = −0.98, *p* = 0.326; *τ* = 0.03, *p* = 0.870) and the correlational subset (*z* = −1.11, *p* = 0.266; *τ* = −0.24, *p* = 0.182). The trim-and-fill test also yielded no additional need for further studies on the left side for the mean difference subset, but it did indicate the need for two additional studies on the *right* side for the correlational subset. It can therefore be assumed that some studies with positive mediocre or large effects within the correlational subset are missing; this is unlike an ordinary publication bias, where negative or unfavorable effects usually remain unpublished [117]. Funnel plots for the two subsets can be found in Figure A5 and Figure A6 in Appendix B. Further, the funnel plot with additional studies after the trim-and-fill test for the correlation subset can be found in Figure A7 in Appendix B. In sum, a publication bias does not seem to be an important issue in the current meta-analysis.

## 4. Discussion

In the current meta-analysis, the overall effect size was significantly positive yet diminutively small. Regarding the conventions for Cohen’s *d* values, an effect size with *d* = 0.2 upwards can be considered small [113], which means that the found effect cannot even be regarded as small. However, the results suggest that people with diagnosed BD and people with higher subclinical scores related to BD indeed show marginally better performance in DT tasks. An important implication of this finding is that BD does not exclusively come with impairments; instead, some areas such as creative thinking might even be slightly improved, yet not on a grand scale. However, the improvement of DT is certainly not as drastic as suggested by the mad genius theory; therefore, having BD should not be seen as something desirable that comes along with a genius level of creative thinking. In fact, the difference in DT performance between healthy and currently manic individuals as well as between euthymic and currently manic individuals was small and not significant, indicating that DT performance is not significantly better during mania. Further, DT performance during a depressive episode was even worse compared to people with a healthy or a euthymic status, which indicates that BD may even temporarily dampen DT performance. Hence, in sum, benefits of BD for creative thinking are practically very limited.

Some other interesting results were found in the moderator analyses. Considering the study and sample level, the only significant moderator was the country in which the study was conducted. Here, one study conducted in Turkey (by Alici et al. [9]) differed from the others in showing a relatively large, negative effect size. When analysing influential studies, this particular study was also found to appear striking in the Baujat plot. Even though cultural differences cannot be ruled out, another reason for this study to stand out may be because this study was the only one to use a neuroimaging paradigm. While all other studies implemented DT assessments in the usual manner using pen-and-paper versions, Alici and colleagues [9] used functional near-infrared spectroscopy to simultaneously measure brain activity. However, a mechanism based solely on the laboratory setting of the study design appears to not be the most reasonable from a theoretical point of view. Hence, sound explanations for this outlier study seem to be rather hard to find.

Regarding the moderators on the level of DT and BD, the DT-related moderators as studied in this work did not seem to be of special significance, while moderators for BD yielded some interesting findings. Regarding DT, the only variable with a significant influence was the classification of *mixed* regarding the DT domain, which was associated with a slightly stronger positive association between BD and DT. Most studies classified as mixed used the ATTA, and thus merged the verbal and figural domains of the TTCT, or used total creativity scores which may have increased the reliability of the DT assessment. However, the explanatory power of a mixed domain is questionable since it can be seen as an umbrella term for studies using non-distinct or confounding categorization of their DT tasks. Unfortunately, no difference between the verbal and the figural domains was found, which would have been of greater interest than the mixed domain. It would further be interesting whether the BD–DT relationship may be moderated by the type of originality scoring. It was proposed that mental illness may especially related to uncommon responses as measured with uniqueness scores but maybe less so to creative responses as assessed by creativity ratings [118]. This question may be addressed in future research when more studies with these different DT originality scorings are available.

Regarding BD moderators, a handful of variables seem to be of importance: First, only BD presentation as subclinical as opposed to bipolar I and bipolar mixed showed a significant positive effect. These findings are consistent with the notion of a U-shaped association between psychopathology and creativity [86], as only subclinical expressions of BD, located in the middle of the continuum of psychopathological severity between healthy and clinical illness, were associated with higher creativity. This corroborates the notion that mild deviations of thought processes can support creative thinking, whereas full-blown clinical psychopathologies are usually associated with mental impairments.

Moreover, BD status as either euthymic, subclinical, depressed, or manic had a moderating effect, indicating that a patient’s current affective episode might influence the strength of the effect, with especially depressive episodes implying reduced creative potential. Further, non-clinical assessments of BD explained a significant amount of variance as well as the HPS assessment tool for BD, in contrast to the *DSM* and others. This might be explained by the intercorrelation of the moderators, since correlational studies mainly made use of the HPS as an assessment tool, and this approach can further be classified as non-clinical.

Another striking finding of the present meta-analysis is the fact that empirical research on the relationship between DT and BD is still scarce: even though we conducted a thorough literature search with over 6000 hits, in the end no more than 13 articles fit the research subject and met the inclusion criteria. Although strict inclusion criteria were applied, which might have reduced the total number of studies, these inclusion criteria were necessary to assure a certain level of empirical quality. Given the small number of included studies, the estimated effect size has to be interpreted with caution, and the implications of the moderator analyses are also limited. Hence, there is a need for a larger body of work surrounding creativity and BD, and future research can consider several aspects to improve study quality. For instance, severity, duration of the illness, onset, number of episodes, hospitalization, and detailed information on patients’ medication were rarely reported, which is why these moderators could not be included and analysed in this meta-analysis. Only one of the five studies examining mean differences gave extensive information on patients’ history with BD, listing the mean age during the first manic and depressive episodes, number of manic and depressive episodes, information on hospitalization due to manic or depressive episodes as well as the specific dosage of various medications [119]. Further, only one study systematically examined the differences in DT during an affective episode as opposed to euthymic affective states: Rybakowski and Klonowska [120] measured DT twice in their sample, once during a manic or depressive episode and once in remission after a manic or depressive episode. They found that DT performance was significantly lower during depressive episodes compared to manic episodes and compared to remission. This allows for a more detailed insight, making clear that BD is far too complex to only compare bipolar patients to healthy controls; one should also examine the changes across different episodes of BD, or at least consider the current episode status.

Another issue that came up regarding study quality was the assessment of DT. Even though all studies oriented their DT scoring to the four scorings of fluency, flexibility, originality, and elaboration, as suggested by Guilford [15,16], calculations of these scorings were rather discordant. Due to high intercorrelations of the four scorings, especially between fluency and originality [22], the scorings are heavily confounded. Confounding between fluency and originality can have a conceptual reason (the equal-odds rule assumes that there is a greater likelihood for original ideas when there is a greater number of overall ideas; [56,58]), but it is commonly just due to summative scoring across responses, which makes all DT facets necessarily dependent on idea fluency. Indeed, a handful of studies did not elaborate on their method of calculating originality, and over half of the studies chose to calculate originality by simply summing all original ideas. Several scoring methods have been proposed to avoid a necessary fluency confound [39] including the scoring [22], snapshot scoring [25], or top-scoring [26]. Average scoring was conducted for only seven effect sizes from four studies, but this should not only be more commonly applied when investigating creativity and BD, but it should be used more generally in all areas of creativity research.

The same issue arose for studies using a *total* score, where either the assessment of DT domains was not specified at all or sub-domains were simply added up together into a total score for creativity. The problem with this type of total score also lies in the high intercorrelations between DT domains since the explanatory power of such a total score remains obscure. As fluency is the number of ideas, flexibility is the number of categories of ideas, and originality is the uniqueness and remoteness of ideas, what is the meaning of the total score? In this meta-analysis, the total score was used as the only scoring method for DT in five studies with eight effect sizes; these studies did not report individual sub-scores for fluency, flexibility, originality, or elaboration. This undermines a better understanding of how BD relates to quantitative versus qualitative facets of creative ideation. Another two studies with five effect sizes reported total scores in addition to scorings of the individual facets. Hence, in sum over half the studies used total scores as a scoring type. The current meta-analysis did control for the use of total scorings beforehand, since analyses were also calculated with a subsample of effect sizes in which additional total scores were excluded; however, their exclusion did not yield any remarkable differences. Thus, for simplicity, the analyses were performed using all the extracted effect sizes. Overall, though, given that the quality of DT assessment tended to be problematic, since confounded methods were used for total scores, the mixed domain, and summation indices, it is hardly surprising that no significant moderators for DT assessment were observed.

Even though the main focus here is to examine the studies included in this meta-analysis, it also makes sense to examine studies that were not included: some studies were excluded for not meeting the inclusion criteria but can still add to the discussion on the relationship between DT and BD. For example, Ghadirian and colleagues [121] did not meet the inclusion criterion of having a healthy CG, but they had a noteworthy approach by comparing bipolar patients to patients with other psychopathologies. The study accounted for the severity of the illness, which was neglected by most of the included studies. Even though the difference in DT scores between bipolar patients and patients with other psychopathologies was not significant, severity seems to play an important role in DT performance [121]. Interestingly, moderately ill patients showed the best creativity scores, even better than mildly ill or recovered patients, while severely ill patients showed the worst performance. These results are again in line with the idea of an inverted U-shaped relationship between psychopathology and creativity, which has often been put forward in the literature [86,87,88]. However, more studies that make use of specific diagnostic instruments to measure severity of BD are needed to better understand the role of severity in DT performance.

### Limitations

Even though the current meta-analysis makes an important contribution to the more elaborated research on a possible psychopathology–creativity link, specifically on a link between BD and DT, it has some limitations. First, not all studies on the relevant topic were obtainable at the time this meta-analysis was conducted, which might have resulted in an incomplete sample of studies. Most studies that could not be obtained were unpublished papers or doctoral dissertations. Even though this meta-analysis showed no indication for a publication bias, including more unpublished studies would have helped obtain a full picture of all existing studies. Further, despite contacting the authors, three studies could not be included due to the lack of an effect size. Including these studies would have strengthened the statistical and informational power of the current meta-analysis, as the final number of 13 studies is rather small.

Second, interpretational insights were limited as a result of unevenly distributed and highly confounded moderators. Noticeably, most moderators were strongly associated with each other, revealing covariations in study design factors, but they may be partly overestimated due to a small number of included studies and effect sizes, as well as infrequent and missing data on moderator presentations. Some presentations of moderators were represented fewer than four times, making the interpretation of their moderating effects rather difficult [122]. More precisely, this concerned the presentations *England* and *Turkey* in the moderator country, since only three effect sizes from England and one from Turkey were present; the presentations *school children* and *mixed* in the moderator age group, since the two presentations were only represented with one effect size each; the presentations *flexibility* and *elaboration* in the moderator DT scoring, since only one effect size was included for each; as well as the presentation *other* in the moderator DT calculation method, which had only two effect sizes. Furthermore, for some moderators, studies generally did not provide a great amount of information, which resulted in a lot of missing values within these moderators. Of concern were the moderators comorbidity and medication since only 5 of the 13 studies reported information on these 2 moderators. Taken together, this might possibly result in higher intercorrelations between moderators.

Additionally, some intercorrelations can also be explained by the characteristics of the studies and the moderators themselves. All included studies were relatively similar in terms of the number of participants, with similar mean age and a similar ratio of male and female participants. Furthermore, study designs in the included studies can all be regarded as quite similar because most of the studies made use of similar assessment tools such as the *DSM* or HPS for BD and the AUT or TTCT for DT. Therefore, it makes sense that other moderators related to the assessment tool are also associated with one another, such as DT time, DT domain, or DT instruction being related to the DT test.

Another similar aspect to consider, which was mentioned before, is the fact that this meta-analysis included both clinical and subclinical samples. When taking a closer look at this, interrelationships also make sense: all the subclinical studies used correlational designs, and all the correlational studies used non-clinical assessment methods in the form of BD questionnaires, mainly the HPS. Therefore, it comes as no surprise that a correlational design was associated with a subclinical presentation of BD, with a BD status as healthy, with a BD assessment tool using the HPS or other questionnaires, as well as with no information on comorbidities and medication. Certain converse associations were also apparent, then, for clinical studies, as most of the clinical studies used mean differences (apart from one that reported both mean differences and correlations). The studies using mean differences usually also made use of clinical assessment tools for BD, such as the *DSM*, had BD presentations as either bipolar I or mixed, had BD status as either euthymic, manic, or depressed, as well as had information on comorbidities and medication. Taken together, intercorrelations between moderators make sense and are also commonly reported as a problem of meta-analyses [114]. Nonetheless, given the high intercorrelations between moderators, findings from moderator analyses need to be interpreted with caution.

Third, the current meta-analysis was not able to directly test predictions regarding an inverted U-shaped relationship between DT and BD. By separating clinical and subclinical samples this study made a first attempt to map different levels of impairment to investigate the shape of the relationship, but it is questionable whether solely separating between clinical and subclinical samples is sufficient to illustrate the whole spectrum of impairment. Nonetheless, studies from correlational designs, which represented subclinical studies, showed larger effect sizes than studies comparing mean differences, which represented clinical studies. This alone is not sufficient to illustrate an inverted U-shaped relationship between impairment due to BD and performance in DT, but it does indicate that subclinical samples, which can be considered to have less impairment, tended to show better performance on DT than clinical samples. It would have been of great interest to use severity as a variable to parametrically investigate differences in DT performance as a function of severity levels of BD.

Even though including clinical and subclinical samples was insightful, using both clinical and subclinical studies could also be considered a limitation, since a full-blown clinical BD diagnosis is not equivalent to heightened subclinical scores on BD-related questionnaires. However, the number of studies would have been even smaller if only clinical studies were included, making the results from this meta-analysis and the moderator analyses less meaningful. Importantly, most correlational studies used diagnostic instruments based on the HPS, which has been found to correlate highly with current hypomanic symptoms [91]. Therefore, it can be assumed that both the clinical and subclinical studies measured the same construct in regard to BD.

Nonetheless, to control for this crucial difference in the study samples (subclinical vs. clinical), we conducted moderator analyses for study design as either correlational (subclinical samples) or mean differences (clinical samples). The moderator analysis of study design yielded no significant differences for the overall moderator, but it showed that a significant BD–DT effect was only observed in subclinical studies. We therefore cannot rule out that differences between clinical and subclinical samples existed but remain obscured due to missing power.

## 5. Conclusions

Despite the limitations listed above, the current meta-analysis helps to shed light onto the relationship between BD and divergent thinking as an indicator of creative potential. This was the first meta-analysis to date that used a theoretical framework other than the “mad genius” hypothesis as a basis to assume a relationship between DT and BD in one direction or the other. Further, it is the first meta-analysis to date that focused on BD exclusively, linking it to little-c creativity as measured by DT, without pooling them with related but different constructs such as convergent thinking creativity or verbal fluency. The results generally indicate that individuals with diagnosed BD (outside of a depressive episode) or higher subclinical ratings of BD indeed perform slightly better on DT tasks, indicating increased creative potential, but not to the magnitude that is sometimes postulated. Even though the small number of studies and the qualitative deficits of some studies limited statistical possibilities, the moderator analyses also indicate that the relationship might be even more complex, since some moderators on the study and sample level as well as moderators around BD had moderating effects.

Overall, this meta-analysis could not identify a succinct “yes-or-no” answer to the question of whether people with BD are more capable in divergent ideation than healthy people. Even though the overall effect size in the current meta-analysis indicates that individuals with BD might indeed have a slight advantage (if any) in DT, it still underlines the need for future quality studies on the link between BD and creativity.

## Figures and Tables

**Figure 1 ijerph-20-06264-f001:**
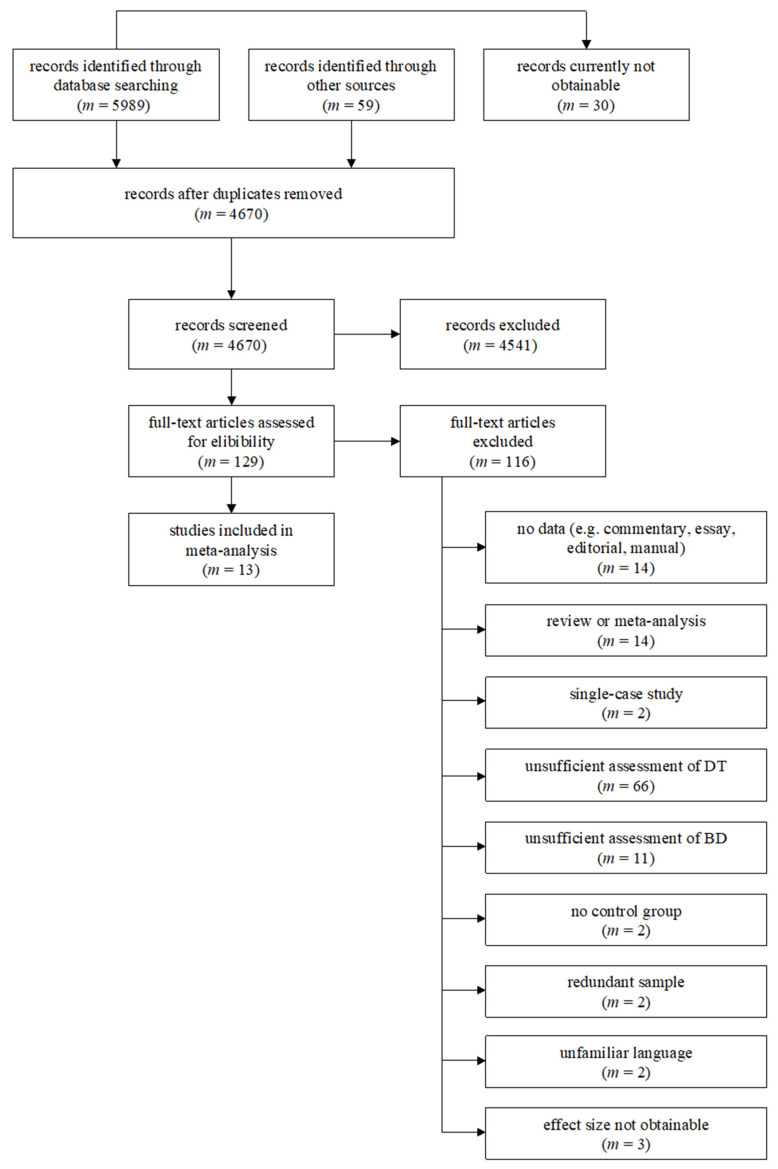
PRISMA flow diagram [107] illustrating the selection process.

**Figure 2 ijerph-20-06264-f002:**
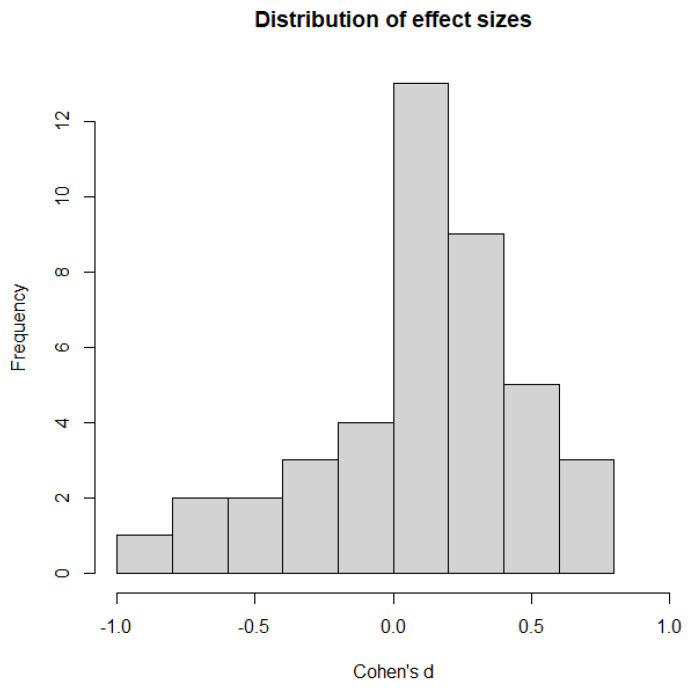
Histogram illustrating the distribution of effect sizes using Cohen’s *d* values [113].

**Figure 3 ijerph-20-06264-f003:**
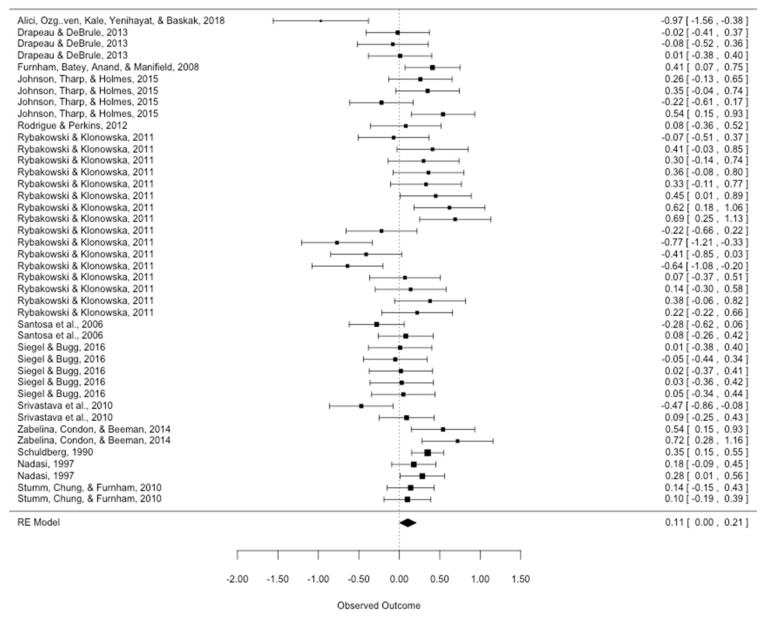
Forest plot for the overall effect including all 42 effect sizes from 13 studies. Study labels on the left side in the forest plot refer to authors and year of publication. If the same study labels appear multiple times this means that multiple effect sizes were obtained from such studies. The file “References_of_studies_included_in_the_meta_analysis.pdf” which includes all references mentioned in this figure and details related to such multiple effect sizes are accessible in the open material for this work: https://osf.io/tnksa/ (accessed on 12 June 2023).

**Figure 4 ijerph-20-06264-f004:**
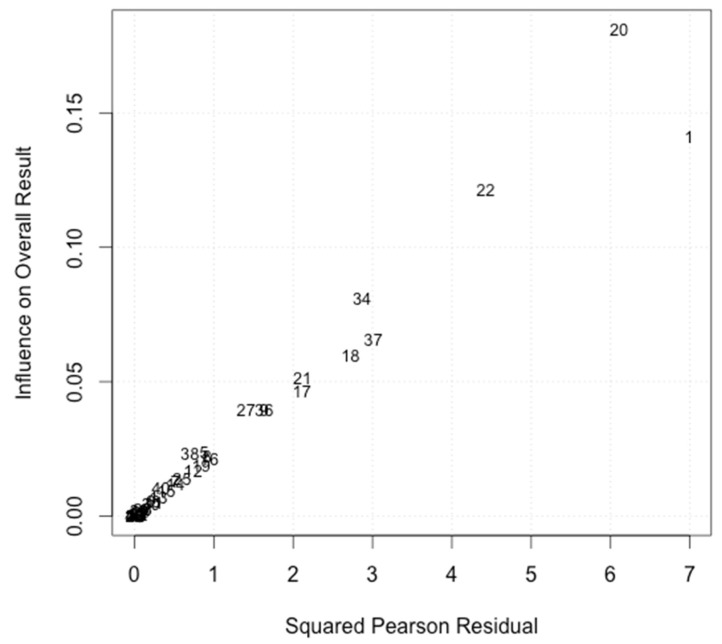
Baujat plot [115] to identify influential studies. Vertical axis shows contribution to the overall effect, and horizontal axis shows heterogeneity.

**Figure 5 ijerph-20-06264-f005:**
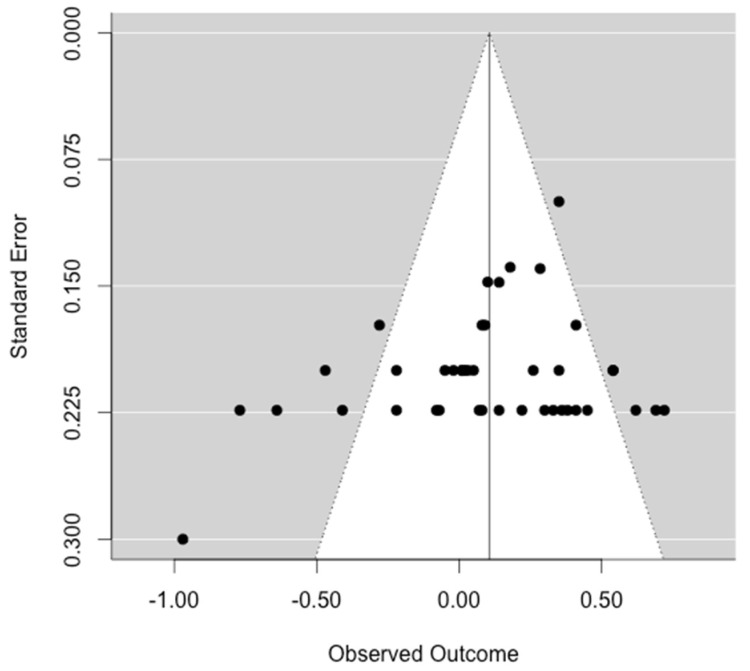
Funnel plot for the overall effect size for analysing publication bias.

## Data Availability

The literature search documentation, data, and analysis script for this meta-analysis are openly available at https://osf.io/tnksa/ (accessed on 12 June 2023) and this meta-analysis was not preregistered.

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
