# Peer review of "The Manic Idea Creator? A Review and Meta-Analysis of the Relationship between Bipolar Disorder and Creative Cognitive Potential"

_ijerph, 2023, doi:10.3390/ijerph20136264_

Round 1

Reviewer 1 Report

Thank you for the opportunity to read this intriguing work on the association between bipolar disorder and creative potential. Overall, I believe the entire work has a courageous dimension, and the paper needs minor revisions. Below are my suggestions about the manuscript.
First, I suggest reducing the abstract length so that readers can understand and get the core of the work faster. Second, whereas the “Creativity, Creative Potential, and Divergent Thinking”, “Bipolar Disorder”, and “Mad Genius Theory” sections are absolutely clear and interesting, in the further paragraph, the section “Shared phenotype” should be improved in order to ensure readability. Third, I suggest authors include a separate section with the limits of the study. Finally, note that I have found some typos throughout the manuscript; I advise the authors to check them.

Author Response

Reviewer #1:

Thank you for the opportunity to read this intriguing work on the association between bipolar disorder and creative potential. Overall, I believe the entire work has a courageous dimension, and the paper needs minor revisions. Below are my suggestions about the manuscript.

Author Response: Thank you for the positive reception of our work. As you can see below, we have addressed each of the issues raised.

First, I suggest reducing the abstract length so that readers can understand and get the core of the work faster.

Author Response: This is a good idea – thank you. We have snow streamlined the abstract to make it more quickly accessible for the reader.

Second, whereas the “Creativity, Creative Potential, and Divergent Thinking”, “Bipolar Disorder”, and “Mad Genius Theory” sections are absolutely clear and interesting, in the further paragraph, the section “Shared phenotype” should be improved in order to ensure readability.

Author Response: We have rewritten the subsection on “Shared phenotype” and found it to be improved particularly towards the end of the section. The content here was quite complex and we tried to use much simpler language now for a clearer presentation.

Third, I suggest authors include a separate section with the limits of the study.

Author Response: This is again a nice suggestion that results in a clearer structure of the paper and we have amended the discussion accordingly.

Finally, note that I have found some typos throughout the manuscript; I advise the authors to check them.

Author Response: We carefully read the paper again and corrected all identified typos. Thank you for your careful reading.

Reviewer 2 Report

The presented manuscript concerns an interesting and potentially important issue, which is an attempt to verify the "myth" about the relationship between bipolar affective disorder (BP) and creative potential. Unfortunately, the form of the manuscript is completely unacceptable because:

- the manuscript as a whole is absurdly long and at the same time very chaotic. No one will read this text entirely because it is a jumble of scientific concepts, philosophical views, and popular opinions. In fact, the goal that the authors set themselves can be achieved with a volume that is 70% shorter.

- the basic substantive error is the use of very complex concepts (creativity), which have an innumerable number of contexts, from very humanistic to quite strict, but these contexts and understandings are often mistaken by the authors.

- the effect achieved, i.e. the lack of an unambiguous decision as to whether BP is related to creative potential may be largely the result of theoretical errors and definitional chaos and not the actual result of empirical work.

- the second fundamental error is that the authors do not distinguish measures of creative processes from mere measurement using standard cognitive tests. In fact, this EXTREMELY LONG manuscript has not developed a final, operational definition of creativity (maybe because there is no such definition due to the fact that it is a humanistic concept, and not from the field of empirical sciences, including clinical science), which would allow the necessary narrowing of the analyzes . What was the final quantitative, quantifiable indicator of the creativity of the respondents?

- far too many abbreviations in the Introduction that make it hard to understand the narratives and get overwhelmed by different concepts

- many concepts and indicators of creative potential overlap with well-known concepts of cognitive functions. The authors do not distinguish between them (e.g. fluence) and generally give new meaning to constructs that have been well described and long measured.

- the concepts of philosophers from 2,000 years ago or even earlier are not a good source of inspiration for modern clinical research.

- verbal fluency is not a test of creativity, but of executive functions.

I recommend replacing the word creativity with "divergent thinking", identifying unambiguous indicators of divergent thinking, showing how they differ from typical measures of executive function (if any), identifying only articles that have actually studied these measures in the BD group, and write a manuscript for 5000 words. In this version, the manuscript is completely unacceptable due to the fact that it is essentially uncommunicative and ultimately inconclusive.

Author Response

Reviewer #2:

The presented manuscript concerns an interesting and potentially important issue, which is an attempt to verify the "myth" about the relationship between bipolar affective disorder (BP) and creative potential. Unfortunately, the form of the manuscript is completely unacceptable because:

Author Response: Thank you for your constructive criticism of our work. As you can see below, we have worked hard on the paper to address each of the issues raised and we are convinced that a stronger paper resulted from the revisions made.

- the manuscript as a whole is absurdly long and at the same time very chaotic. No one will read this text entirely because it is a jumble of scientific concepts, philosophical views, and popular opinions. In fact, the goal that the authors set themselves can be achieved with a volume that is 70% shorter.

Author Response: We have streamlined the paper in all parts. Thank you for pointing out that streamlining was indeed needed for a clearer presentation of our work.

- the basic substantive error is the use of very complex concepts (creativity), which have an innumerable number of contexts, from very humanistic to quite strict, but these contexts and understandings are often mistaken by the authors.

Author Response: Creativity is indeed a complex construct and even creativity researchers do not necessarily agree on the very definition of creativity. Hence, we decided to stuck with the standard definition and move theoretically to the more broadly applicable concept of creative potential and the related idea of everyday creativity. Divergent thinking tasks are the most widely used assessments for both of these creativity-related concepts. Hence, we agree with your comment (and also the comment below) and have sharpened this overall conceptualization of creativity in our work. Specifically, we have now…

  • …presented the standard definition of creativity first and move then quickly to divergent thinking as an indicator of creative potential and everyday creativity.
  • …changed instances of the word “creativity” into “DT” whenever the latter term was more appropriate.
  • …renamed sections into names that emphasize a focus on DT rather than creativity and streamlined the content of subsections accordingly.

- the effect achieved, i.e. the lack of an unambiguous decision as to whether BP is related to creative potential may be largely the result of theoretical errors and definitional chaos and not the actual result of empirical work.

Response: Our work is one of the most systematic meta-analytical investigations of the relationship between BD and creativity. Our unique focus on DT in this regard emphasizes the contribution of our work. Recent meta-analyses in the field of creativity research has strongly emphasized the need to analyze assessment-related moderators which is particularly important for DT assessment. We have emphasized this now on p. 3, lines 98-100.

- the second fundamental error is that the authors do not distinguish measures of creative processes from mere measurement using standard cognitive tests. In fact, this EXTREMELY LONG manuscript has not developed a final, operational definition of creativity (maybe because there is no such definition due to the fact that it is a humanistic concept, and not from the field of empirical sciences, including clinical science), which would allow the necessary narrowing of the analyzes . What was the final quantitative, quantifiable indicator of the creativity of the respondents?

Author Response: As explained above, we have now sharpened the used definitions.

- far too many abbreviations in the Introduction that make it hard to understand the narratives and get overwhelmed by different concepts

Author: We have now removed abbreviations that are only rarely used to increase the readability of the work.

- many concepts and indicators of creative potential overlap with well-known concepts of cognitive functions. The authors do not distinguish between them (e.g. fluence) and generally give new meaning to constructs that have been well described and long measured.

Author Response: We have carefully checked the paper in this regard. Unfortunately, we were not able to find the mentioned parts in the manuscript. However, we would be happy to revise the paper further in case the reviewer would point us more concretely to the parts that require revisions.

- the concepts of philosophers from 2,000 years ago or even earlier are not a good source of inspiration for modern clinical research.

Author Response: We removed the mentioned content from the paper.

- verbal fluency is not a test of creativity, but of executive functions.

 Author Response: Thank you for this insightful comment. We couldn’t agree more. However, we did not find a single sentence in which we equate verbal fluency with a test of creativity. Actually, we refer to others who did exactly that and argue that we do not follow such a loose conceptualization (see p. 12, lines 595 to 601).

I recommend replacing the word creativity with "divergent thinking", identifying unambiguous indicators of divergent thinking, showing how they differ from typical measures of executive function (if any), identifying only articles that have actually studied these measures in the BD group, and write a manuscript for 5000 words.

Author Response: Thank you for your feedback. This is exactly what we did.

In this version, the manuscript is completely unacceptable due to the fact that it is essentially uncommunicative and ultimately inconclusive.

Author Response: We sincerely hope that we could clarify some misunderstandings that emerged perhaps from not clearly communicating all the concepts in our work.

Reviewer 3 Report

The authors conducted a meta-analysis to review the relationship between Bipolar Disorder and Creative Cognitive potential. To determine the relationship, a random-effects meta-analytical model was used to account for effect size variations, and meta-regression models were investigated to explain these variations. In the present study, the authors found an association between divergent thinking and bipolar disorder. It is well written, presented, and discussed appropriately. Therefore, the manuscript can be accepted in its present form.

However, there is room for improvement.

For example - Variables "connected" to Bipolar use "associated."

Author Response

Reviewer #3

The authors conducted a meta-analysis to review the relationship between Bipolar Disorder and Creative Cognitive potential. To determine the relationship, a random-effects meta-analytical model was used to account for effect size variations, and meta-regression models were investigated to explain these variations. In the present study, the authors found an association between divergent thinking and bipolar disorder. It is well written, presented, and discussed appropriately. Therefore, the manuscript can be accepted in its present form.

However, there is room for improvement.

Author Response: Thank you for the positive reception of our work. We have greatly streamlined the paper wherever possible and are convinced that a more concise paper resulted from this review process. 

For example - Variables "connected" to Bipolar use "associated."

Author Response: Thank you for pointing us to this issue. We have amended all instances of “connected” into “associated”.